# Joint-task Self-supervised Learning
# for Temporal Correspondence

**Xueting Li**[1]*, **Sifei Liu**[2]*, **Shalini De Mello**[2], **Xiaolong Wang**[3], **Jan Kautz**[2], **Ming-Hsuan Yang**[1]
[1]University of California, Merced, [2]NVIDIA, [3] Carnegie Mellon University

## Abstract

This paper proposes to learn reliable dense correspondence from videos in a self-supervised manner. Our learning process integrates two highly related tasks: tracking large image regions *and* establishing fine-grained pixel-level associations between consecutive video frames. We exploit the synergy between both tasks through a shared inter-frame affinity matrix, which simultaneously models transitions between video frames at both the region- and pixel-levels. While region-level localization helps reduce ambiguities in fine-grained matching by narrowing down search regions; fine-grained matching provides bottom-up features to facilitate region-level localization. Our method outperforms the state-of-the-art self-supervised methods on a variety of visual correspondence tasks, including video-object and part-segmentation propagation, keypoint tracking, and object tracking. Our self-supervised method even surpasses the fully-supervised affinity feature representation obtained from a ResNet-18 pre-trained on the ImageNet. The project website can be found at https://sites.google.com/view/uvc2019/.

## 1   Introduction

Learning representations for visual correspondence is a fundamental problem that is closely related to a variety of vision tasks: correspondences between multi-view images relate 2D and 3D representations, and those between frames link static images to dynamic scenes. To learn correspondences across frames in a video, numerous methods have been developed from two perspectives: (a) learning region/object-level correspondences, via object tracking [2, 42, 44, 37, 49] or (b) learning pixel-level correspondences between multi-view images or frames, e.g., via stereo matching [35] or optical flow estimation [29, 41, 16, 31].

However, most methods address one or the other problem and significantly less effort has been made to solve both of them together. The main reason is that methods designed to address either of them optimize different goals. Object tracking focuses on learning object representations that are invariant to viewpoint and deformation changes, while learning pixel-level correspondence focuses on modeling detailed changes within an object over time. Subsequently, the existing supervised methods for these two problems often use different annotations. For example, bounding boxes are annotated in real videos for object tracking [53]; and pixel-wise associations are generated from synthesized data for optical flow estimation [4, 10]. Datasets with annotations for both tasks are scarcely available and supervision, here, is a further bottleneck preventing us from connecting the two tasks.

In this paper, we demonstrate that these two tasks inherently require the same operation of learning an inter-frame transformation that associates the contents of two images. We show that the two tasks benefit greatly by modeling them jointly via a single transformation operation which can simultaneously match regions and pixels. To overcome the lack of data with annotations for both tasks we exploit self-supervision via the signals of (a) *Temporal Coherency*, which states that objects or scenes move smoothly and gradually over time; (b) *Cycle Consistency*, correct correspondences

---

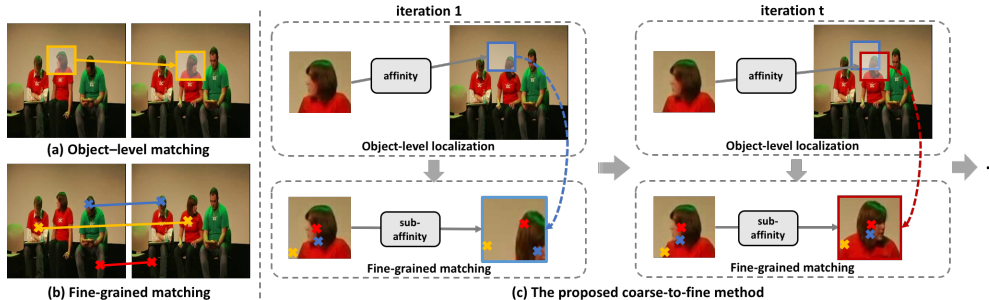

**iteration 1**       **iteration t**

affinity    Object-level localization    affinity    Object-level localization

sub-affinity    Fine-grained matching    sub-affinity    Fine-grained matching

(a) Object–level matching    (b) Fine-grained matching    (c) The proposed coarse-to-fine method

Figure 1: Our method (c) compared against (a) region-level matching (e.g., object tracking), and (b) pixel-level matching, e.g., matching by colorization [45]. We propose a joint-task framework which conducts region-level and fine-grained matching simultaneously and which are supported by a single inter-frame affinity matrix $A$. During training, the two tasks improve each other progressively. To illustrate this, we unroll two training iterations and illustrate the improvement with the red box and arrow.

should ensure that pixels or regions match bi-directionally and (c) *Energy Preservation*, which preserves the energy of feature representations during transformations. Since all these supervisory signals naturally exist in videos and are task-agnostic, the transformation that we learn through them can generalize well to any video without restriction on domain or object category.

Our key idea is to learn *a single* affinity matrix for modeling *all* inter-frame transformations through a network that learns appropriate feature representations that model the affinity. We show that region localization and fine-grained matching can be carried out by sharing the affinity in a fully differentiable manner: the region localization module finds a pair of patches with matching parts in the two frames (Figure 1, mid-top), and the fine-grained module reconstructs the color feature by transforming it between the patches (Figure 1, mid-bottom), all through the same affinity matrix. These two tasks symbiotically facilitate each other: the fine-grained matching module learns better feature representations that lead to an improved affinity matrix, which in turn generates better localization that reduces the search space and ambiguities for fine-grained matching (Figure 1, right).

The contributions of this work are summarized as: (a) A joint-task self-supervision network is introduced to find accurate correspondences at different levels across video frames. (b) A general inter-frame transformation is proposed to support both tasks and to satisfy various video constraints – coherency, cycle, and energy consistency. (c) Our method outperforms state-of-the-art methods on a variety of visual correspondence tasks, e.g., video instance and part segmentation, keypoints tracking, and object tracking. Our self-supervised method even surpasses the fully-supervised affinity feature representation obtained from a ResNet-18 pre-trained on the ImageNet [9].

## 2 Related Work

Learning correspondence in time is widely explored in visual tracking [2, 42, 44, 37, 49] and optical flow estimation [41, 29, 16]. Existing models are mainly trained on large annotated datasets, which require significant efforts. To overcome the limit of annotations, numerous methods have been developed to learn correspondences in a self-supervised manner [46, 52, 45]. Our work establishes on learning correspondence with self-supervision, and we discuss the most related methods here.

**Object-level correspondence.** The goal of visual tracking is to determine a bounding box in each frame based on an annotated box in the reference image. Most methods belong to one of the two categories that use: (a) the *tracking-by-detection* framework [1, 20, 47, 25], which models tracking as detection applied independently to individual frames; or (b) the *tracking-by-matching* framework that models cross-frame relations and includes several early attempts, e.g., mean-shift trackers [8, 55], kernelized correlation filters (KCF) [14, 27], and several works that model correlation filters as differentiable blocks [32, 33, 7, 48]. Most of these methods use annotated bounding boxes [53] in every frame of the videos to learn feature representations for tracking. Our work can be viewed as exploiting the *tracking-by-matching* framework in a self-supervised manner.

**Fine-grained correspondence.** Dense correspondence between video frames has been widely applied for optical flow and motion estimation [31, 41, 29, 16], where the goal is to track individual pixels. Most deep neural networks [16, 41] are trained with the objective of regressing the ground-truth optical flow produced by synthetic datasets [4, 10]. In contrast to many classic methods [31, 29] that model dense correspondence as a matching problem, direct regression of pixel offsets has limited

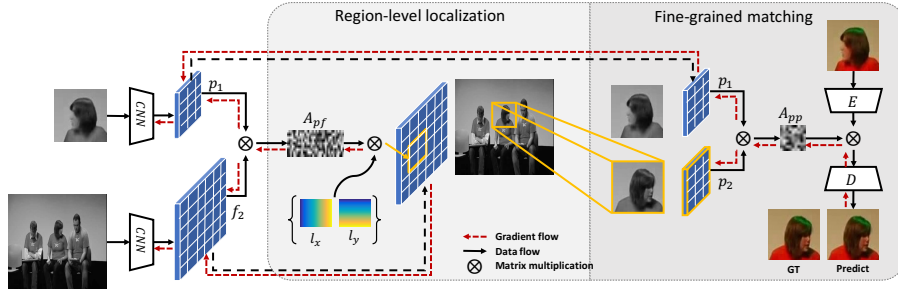

Figure 2: Main steps of proposed method. Blue grids represent the reference-patch $p_1$'s and target-frame $f_2$'s feature maps that are shared by the region-level localization (left box) and fine-grained matching (right box) modules. $A_{pf}$ is the affinity between $p_1$ and $f_2$, and $A_{pp}$ is that between $p_1$ and $p_2$. $p_2$ is a differentiable crop from the frame $f_2$. The maps $l_x$ and $l_y$ are the coordinates of pixels on a regular grid. All modules are differentiable, where the gradient flow is visualized via the red dashed arrows.

capability for frames containing dramatic appearance changes [3, 40], and suffers from problems related to domain shift when applied to real-world scenarios.

**Self-supervised learning.** Recently, numerous approaches have been developed for correspondence learning via various self-supervised signals, including image [17] or color transformation [45] and cycle-consistency [52, 46]. Self-supervised learning of correspondence in videos has been explored along the two different directions – for region-level localization [52, 46] and for fine-grained pixel-level matching [45, 23]. In [46], a correlation filter is learned to track regions via a cycle-consistency constraint, and no pixel-level correspondence is determined. [52] develops patch-level tracking by modeling the similarity transformation of pixels within a fixed rectangular region. Conversely, several methods learn a matching network by transforming color/RGB information between adjacent frames [45, 24, 23]. As no region-level regularization is exploited, these approaches are less effective when color features are less distinctive (see Figure 1(b)). In contrast, our method learns object-level and pixel-level correspondence jointly across video frames in a self-supervised manner.

## 3   Approach

Video frames are temporally coherent in nature. For a pair of adjacent frames, pixels in a later frame can be considered as being copied from some locations of an earlier one with slight appearance changes conforming to object motion. This "copy" operator can be expressed via a linear transformation with a matrix $A$, in which $A_{ij} = 1$ denotes that the pixel $j$ in the second frame is copied from pixel $i$ in the first one. An approximation of $A$ is the inter-frame affinity matrix [44, 30, 52]:

$$A_{ij} = \kappa(f_{1i}, f_{2j}) \tag{1}$$

where $\kappa$ denotes some similarity function. Each entry $A_{ij}$ represents the similarity of subspace pixels $i$ and $j$ in the two frames $f_1 \in \mathcal{R}^{C \times N_1}$ and $f_2 \in \mathcal{R}^{C \times N_2}$, where $f \in \mathcal{R}^{C \times N}$ is a vectorized feature map with $C$ channels and $N$ pixels. In this work, our goal is to learn the feature embedding $f$ that optimally associates the contents of the two frames.

One free supervisory signal that we can utilize is color. To learn the inter-frame transformation in a self-supervised manner, we can slightly modify (1) to generate the affinity via features $f$ learned only from gray-scale images. The learned affinity is then utilized to map the color channels from one frame to another [45, 30], while using the ground-truth color as the self-supervisory signal.

One strict assumption of this formulation is that the paired frames need to have the same contents – no new object or scene pixel should emerge over time. Hence, the existing methods [45, 30] sample pairs of frames either uniformly, or randomly within a specified interval, e.g., 50 frames. However, it is difficult to determine a "perfect" interval as video contents may change sporadically. When transforming color from a reference frame to a target one, the objects/scene pixels in the target frame may not exist in the reference frame, thereby leading to wrong matches and an adverse effect on feature learning. Another issue is that a large portion of the video frames are "static", in which the sampled pair of frames are almost the same and cause the learned affinity to be an identity matrix.

We show that the above problems can be addressed by incorporating a region-level localization module. Given a pair of reference and target frames, we first randomly sample a patch in the reference frame and localize this patch in the target frame (see Figure 2). The inter-frame color transformation is

then estimated between the paired patches. Both localization and color transformation are supported by a single affinity derived from a convolutional neural network (CNN) based on the fact that the affinity matrix can simultaneously track locations and transform features discussed in this section.

## 3.1 Transforming Feature and Location via Affinity

We sample a pair of frames and denote the $1^{st}$ frame as the reference and the $2^{nd}$ one as the target. The CNN can be any effective model, e.g., ResNet-18 [13] with the first 4 blocks that takes a gray-scale image as input. We compute the affinity and conduct the feature transformation and localization on the top layer of the CNN, with features that are one-eighth the size of the input image. This ensures the affinity matrix to be memory efficient and each pixel in the feature space to contain considerable local contextual information.

**Transforming feature representations.** We adopt the dot product for $\kappa$ in (1) to compute the affinity, where each column can be interpreted as the similarity score between a point in the target frame to all points in the reference frame. For dense correspondence, the inter-frame affinity needs to be sparse to ensure one-to-one mapping. However, it is challenging to model a sparse matrix in a deep neural network. We relax this constraint and encourage the affinity matrix to be sparse by normalizing each column with the softmax function, so that the similarity score distribution can be peaky and only a few pixels with high similarity in the reference frame are matched to each point in the target frame:

$$A_{ij} = \frac{\exp(f_{1i}^\top f_{2j})}{\sum_k \exp(f_{1k}^\top f_{2j})}, \quad \forall i \in [1, N_1], j \in [1, N_2] \tag{2}$$

where the variable definitions follow (1). The transformation is carried out as $\hat{c}_2 = c_1 A$, where $A \in \mathcal{R}^{N_1 \times N_2}$, and $c_i$ has the same number of entries as $f_i$ and can be features of the reference frame or any associated label, e.g., color, segmentation mask or keypoint heatmap.

**Tracing pixel locations.** We denote $l_j = (x_j, y_j), l \in \mathcal{R}^{2 \times N}$ as the vectorized location map for an image/feature with $N$ pixels. Given a sparse affinity matrix, the location of an individual pixel can be traced from a reference frame to an adjacent target frame:

$$l_j^{12} = \sum_{k=1}^{N_1} l_k^{11} A_{kj}, \quad \forall j \in [1, N_2] \tag{3}$$

where $l_j^{mn}$ represents the coordinate in frame $m$ that transits to the $j^{th}$ pixel in frame $n$. Note that $l^{nn}$ (e.g., $l^{11}$ in (3)) usually represents a canonical grid as shown in Figure 3.

## 3.2 Region-level Localization

In the target frame, region-level localization aims to localize a patch randomly selected from the reference frame by predicting a bounding box (denoted as "bbox") on a region that shares matching parts with the selected patch. In other words, it is a differential region of interest (ROI) with learnable center and scale. We compute an $N_1 \times N_2$ affinity $A_{pf}$ according to (2) between feature representations of the patch in the reference frame, and that of the whole target frame (see Figure 2(a)).

**Locating the center.** To track the center position of the reference patch in the target frame, we first localize each individual pixel of the reference patch $p_1$ in the target frame $f_2$, according to (3). As we obtain the set $l^{21}$, with the same number of entries as $p_1$, that collects the coordinates of the most similar pixels in $f_2$, we can compute the average coordinate $C^{21} = \frac{1}{N_1} \sum_{i=1}^{N_1} l_i^{21}$ of all the points, as the estimated new position of the reference patch.

**Scale modeling.** For region-level tracking, the reference patch may undergo significant scale changes. Scale estimation in object tracking is challenging and existing methods mainly enumerate possible scales [2, 46] and select the optimal one. In contrast, the scale can be estimated by our proposed model. We assume that the transformed locations $l^{21}$ are still distributed uniformly in a local rectangular region. By denoting $w$ as the width of the new bounding box, the scale is estimated by:

$$\hat{w} = \frac{2}{N_1} \sum_{i=1}^{N_1} \left\| x_i - C^{21}(x) \right\|_1 \tag{4}$$

where the $x_i$ is the x-coordinate of the $i^{th}$ entry in the $l^{21}$. We note that (4) can be proved by using the analogous continuous space. Suppose there is a rectangle with scale $(2w, 2h)$ and with its center located at the origin of a 2D coordinate plane. By integrating points inside of it, we have:

$$\frac{1}{w} \int_{-w}^{w} \|x\|_1 \, dx = \frac{2}{w} \int_0^w x \, dx = w \tag{5}$$

This represents the average absolute distances w.r.t. the center when transforming to the discrete space. The estimation of height is conducted in the same manner.

**Moving as a unit.** An important assumption in the aforementioned ROI estimation in the target frame is that the pixels from the reference patch should move in unison – this is true in most videos, as an object or its parts typically move as one unit at the region level. We enforce this constraint with a *concentration* regularization [58, 15] term on the transformed pixels, with a truncated loss to penalize these points from moving too far away from the center:

$$L_c = \begin{cases} 0, & \left\| l_j^{12}(x) - C^{12}(x) \right\|_1 \le w \text{ and } \left\| l_j^{12}(y) - C^{12}(y) \right\|_1 \le h \\ \frac{1}{N_2} \sum_{j=1}^{N_2} \left\| l_j^{12} - C^{12} \right\|_2, & \text{otherwise} \end{cases} \quad (6)$$

This formulation encourages all the tracked pixels, originally from a patch, to be concentrated (see Figure 3) rather than being dispersed to other objects, which is likely to happen for methods that are based on pixel-wise matching only, e.g., when matching by color reconstruction, pixels of different objects having similar colors may match each other, as shown in Figure 1(b).

### 3.3 Fine-grained Matching

Fine-grained matching aims to reconstruct the color information of the located patch in the target frame, given the reference patch (see Figure 1). We re-use the inter-frame affinity $A_{pf}$ by extracting a sub-affinity matrix $A_{pp}$ containing the columns corresponding to the located pixels in the target frame, and by using it for the color transformation described in the formulations in Section 3.1. To make the color feature compatible with the affinity matrix, we train an auto-encoder that learns to reconstruct an image in the Lab

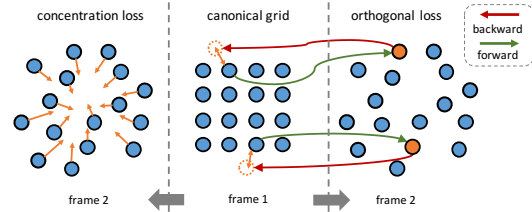

Figure 3: Concentration (left) and orthogonal (right) regularization. The dots denote pixels in feature space. The orange arrows show how they push the pixels.

space faithfully (see the encoder $E$ and the decoder $D$ in Figure 2). This network also encodes global contextual information from color channels. We show that using the color feature instead of pixels significantly reduces the errors caused by reconstructing color directly in the image space [45] (see Table 1, ours vs. [45]). In the following, we introduce self-supervisory signals as regularization for fine-grained matching. For brevity, we denote $A$ as the sub-affinity, $l$ and $f$ as the vectorized coordinate and feature map, respectively, for the paired patches.

**Orthogonal regularization.** Another important constraint, *cycle-consistency*, for the transformation of both location [52] and feature [30] is the orthogonal regularization. For a pair of patches, we encourage every pixel to fall into the same location after one cycle of forward and backward tracking, as shown in Figure 3 (middle and right):

$$l^{\hat{1}2} = l^{11} A_{1 \to 2}, \quad l^{\hat{1}1} = l^{\hat{1}2} A_{2 \to 1} \quad (7)$$

Here we specifically add $m \to n$ to denote affinity transforming from the frame $m$ to $n$, i.e., $A_{m \to n} = \kappa(f_m, f_n)$. Similarly, the *cycle-consistency* can be applied to the feature space:

$$\hat{f}_2 = f_1 A_{1 \to 2}, \quad \hat{f}_1 = \hat{f}_2 A_{2 \to 1} \quad (8)$$

We show that enforcing *cycle-consistency is equivalent to regularizing $A$ to be orthogonal*: With (7) and (8), it is easy to show that the optimal solution is achieved when $A_{1 \to 2}^{-1} = A_{2 \to 1}$. Inspired by recent style transfer methods [12, 30], the color energy represented by the Gram-matrix should be consistent such that $f_1 f_1^\top = f_2 f_2^\top$, which derives that $A_{1 \to 2}^\top = A_{2 \to 1}$ is the goal to reconstruct the color information. Thus, it is easy to show that regularizing $A$ as orthogonal automatically satisfies the cycle constraint. In practice, we switch the role of reference and target to perform the transformation, as described in (7) and (8). We use the MSE loss between both $l^{\hat{1}1}$ and $l^{11}$, $\hat{f}_1$ and $f_1$, and specifically replace $A_{2 \to 1}$ with $A_{1 \to 2}^\top$ in Eq. (8) to enforce the regularization. Namely, the orthogonal regularization provides a concise mathematical formulation for many recent works [52, 46] that exploit *cycle-consistency* in videos.

**Concentration regularization.** We additionally apply the concentration loss (i.e., Eq.(6) without the truncation) in local, non-overlapping $8 \times 8$ grids of a feature map, to encourage local context or

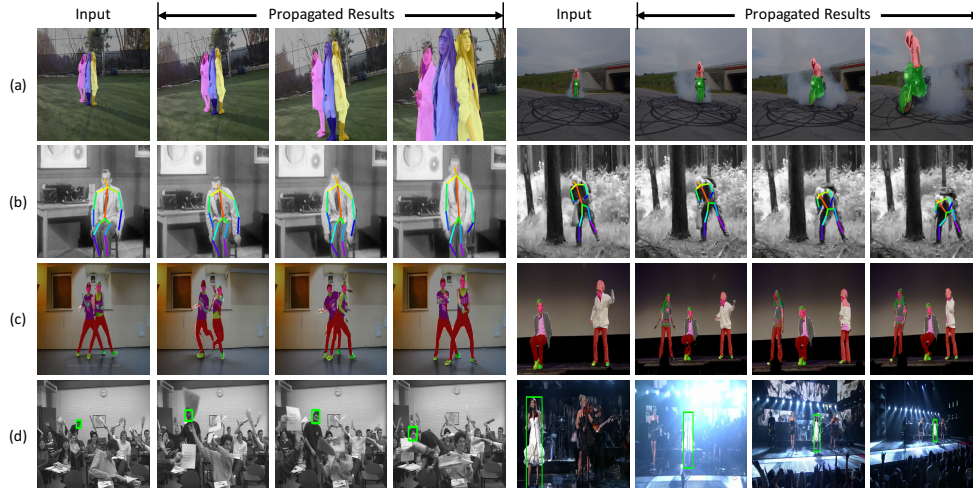

Figure 4: Visualization of the propagation results. (a) Instance mask propagation on the DAVIS-2017 [36] dataset. (b) Pose keypoints propagation on the J-HMDB [19] dataset. (c) Parts segmentation propagation on the VIP [59] dataset. (d) Visual tracking on the OTB2015 [53] dataset.

object parts to move as an entity over time. Unlike [52, 39] where local patches are regularized by similarity transformation via a spatial transformation network [18], this local concentration loss is more flexible by allowing arbitrary deformations within each local grid.

## 4 Experiments

We compare with state-of-the-art algorithms [45, 46, 52] on several tasks: instance mask propagation, pose keypoints tracking, human parts segmentation propagation and visual tracking.

### 4.1 Network Architecture

As shown in Figure 2, our model consists of a region-level localization module and a fine-grained matching module that share a feature representation network (see Figure 2). We use the ResNet-18 [13] as the network for fair comparisons with [45, 52]. The patch randomly cropped from the reference frame is of $256 \times 256$ pixels. We carry out all our experiments on servers equipped with four 16GB Tesla V100 GPUs.

**Training.** We first train the auto-encoder in the matching module (the encoder "E" and decoder "D" in Figure 2) to reconstruct images in the Lab space using the MSCOCO [28] dataset. We then fix it and train the feature representation network using the Kinetics dataset [21]. For all experiments, we train our model from scratch without any level of pre-training or human annotations. The objectives include: (a) concentration loss (Section 3.2 and 3.3), (b) color reconstruction loss and (c) orthogonal regularization (Section 3.3). Involving the localization module from the beginning in the training process prevents the network from converging because poor localization makes matching impossible. Thus we first train our network using patches cropped at the same location with the same size in the reference and target frame respectively. Fine-grained matching is conducted between the two patches for 10 epochs. We then jointly train the localization and matching module for another 10 epochs. We train our model using Adam [22] as the optimizer with a learning rate of $10^{-4}$ for the warm-up and $0.5 \times 10^{-4}$ for the joint training of the localization and matching modules. We set the temperature in the softmax layer applied to the affinity matrix to 1 which empirically achieves best performance.

**Inference.** In the inference stage, we directly apply the affinity learned to transform color feature representations, on different types of inputs, e.g., segmentation masks and keypoint maps. We use the same testing protocol as Wang et al. [52] for all tasks. Similar to [52], we adopt a recurrent inference strategy by propagating the ground truth segmentation mask or keypoint heatmap from the first frame, as well as the predicted results from the preceding $n$ frames onto the target frame. We average all $n + 1$ predictions to obtain the final propagated map ($n$ is 1 for the VIP, and 7 for all the other tasks). For fair comparisons, we also use the k-NN propagation schema as Wang et al. [52] and set $k = 5$ for all tasks. To compare with the ResNet-18 trained on the ImageNet with classification labels, we replace our learned network weights with it and leave other settings unchanged for fair comparisons.

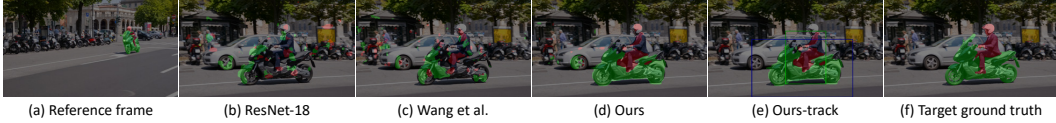

| (a) Reference frame | (b) ResNet-18 | (c) Wang et al. | (d) Ours | (e) Ours-track | (f) Target ground truth |

Figure 5: Qualitative comparison with other methods. (a) Reference frame with instance masks. (b) Results by the ResNet-18 trained on ImageNet. (c) Results by Wang et al. [52]. (d) Ours (global matching). (e) Ours with localization during inference. (f) Target frame with ground truth instance masks.

Table 1: Evaluation of instance segmentation propagation on the DAVIS-2017 dataset [36].

| Model | Supervised | Dataset | $\mathcal{J}$(Mean) | $\mathcal{J}$(Recall) | $\mathcal{F}$(Mean) | $\mathcal{F}$(Recall) |
|---|---|---|---|---|---|---|
| SIFT Flow [29] | × | - | 33.0 | - | 35.0 | - |
| DeepCluster [6] | × | YFCC100M [43] | 37.5 | - | 33.2 | - |
| Transitive Inv [51] | × | - | 32.0 | - | 26.8 | - |
| Vondrick et al. [45] | × | Kinetics [21] | 34.6 | 34.1 | 32.7 | 26.8 |
| Wang et al. [52] ($400 \times 400$) | × | VLOG [11] | 43.0 | 43.7 | 42.6 | 41.3 |
| Wang et al. [52] (480p) | × | VLOG [11] | 46.4 | 50.1 | 50.0 | 48.0 |
| mgPFF [23] | × | - | 42.2 | 41.8 | 46.9 | 44.4 |
| Lai et al. [24] | × | Kinetics [21] | 47.7 | - | 51.3 | - |
| ours | × | Kinetics [21] | 56.8 | 65.7 | 59.5 | 65.1 |
| ours-track | × | Kinetics [21] | **57.7** | **67.1** | **60.0** | **65.7** |
| ResNet-18(3 blocks) | ✓ | ImageNet [9] | 49.4 | 52.9 | 55.1 | 56.6 |
| ResNet-18(4 blocks) | ✓ | ImageNet [9] | 40.2 | 36.1 | 42.5 | 36.6 |
| FlowNet2 [16] | ✓ | FlyingThings3D [34] | 26.7 | - | 25.2 | - |
| PWC-Net [41] | ✓ | FlyingThings3D [34] | 35.2 | 34.0 | 37.4 | 33.1 |
| SiamMask [50] | ✓ | YouTube-VOS [54] | 54.3 | 62.8 | 58.5 | 67.5 |
| OSVOS [5] | ✓ | ImageNet,DAVIS [36] | 56.6 | 63.8 | 63.9 | 73.8 |

## 4.2 Instance Segmentation Mask Propagation on the DAVIS-2017 dataset

Figure 4 (a) and Figure 5 show the propagated instance masks and Table 1 lists quantitative results of all evaluated methods based on the Jacaard index $\mathcal{J}$ (IOU) and contour-based accuracy $\mathcal{F}$. We use the full 480p images during inference for our model. For fair comparisons we test the model by Wang et al. [52] with the resolution of 480p, in addition to the result reported using $400 \times 400$ images. Our model performs favorably against the self-supervised state-of-the-art methods. Specifically, our model outperforms Wang et al. [52] by 13.3% in $\mathcal{J}$ and 16.6% in $\mathcal{F}$. and is even 6.9% better in $\mathcal{J}$ and 4.1% better in $\mathcal{F}$ than the ResNet-18 model [13] trained on ImageNet [9] with classification labels.

Furthermore, we demonstrate that by including the localization module during inference, our model can exclude noise from background pixels. Given the instance masks in the first frame, we obtain the bounding box w.r.t. the instance mask and first locate it in the target frame by our localization module. Then, we propagate the instance masks within the bounding box in the reference frame to the localized bounding box in the target frame using our matching module. Since the propagation is carried out within two bounding boxes instead of the entire frames, we can minimize noise introduced by background pixels as shown in Figure 5 (d) and (e). The quantitative evaluation of this improved model outperforms the model that does not include the localization module during inference. (see "Ours-track" vs. "Ours" in Table 1)

## 4.3 Ablation Studies on the DAVIS-2017 Dataset

We carry out ablation studies to see the contributions of each term, as shown in Figure 6 and Table 2. Note that inference is conducted between a pair of full-size frames without localization.

**Region-level Localization.** Our model trained with the region-level localization module is able to place the individual points all within a reasonable local region (Figure 6 (c)). We show that the model can accurately capture both region-level shifts (e.g., person moving forward), and subtle deformations (e.g., movement of body parts), while preserving the correct spatial relations among all the points. In contrast, the model trained without the localization module tends to model global matching, leading to less accurate preservation of the local spatial relationships among points, e.g., the red points in Figure 6 (d) tend to cluster together as shown in the cyan circle. Consistent quantitative results can also be found in Table 2 (c), where the $\mathcal{J}$ and $\mathcal{F}$ measures drop 2.5% and 0.9%, respectively, when trained without the localization module. We also discover that the localization module should always be trained together with the concentration loss to satisfy the assumption in Section 3.2(Table 2(f)(g)).

**Concentration regularization.** The concentration regularization encourages locality during the transformation process, i.e. points within a neighbourhood in the reference frame stay together in the target frame. The model trained without it tends to introduce outliers, as shown in the cyan circle of

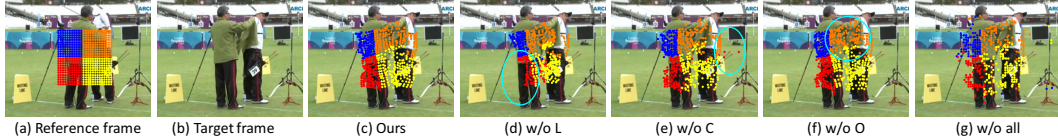

| (a) Reference frame | (b) Target frame | (c) Ours | (d) w/o L | (e) w/o C | (f) w/o O | (g) w/o all |

Figure 6: Visualization of the ablation studies. Given a set of points in the reference frame (a), we visualize the results of propagating these points on to the target frame (b). "L", "C", "O" and "all" correspond to the localization modules, concentration or orthogonal regularization, or all of them (d-g).

Table 2: Ablation studies. The minus sign "-" indicates training without the specific module or regularization. "L", "O" and "C" mean the localization module, orthogonal and concentration regularization, respectively. The last column ("(g) -all") shows results of a baseline model trained without any of "L", "O" or "C".

| Metric | (a) Ours-track | (b) Ours | (c) -L | (d) -O | (e) -C | (f) -O&C | (g) -all |
|---|---|---|---|---|---|---|---|
| $\mathcal{J}$ (Mean) | 57.7 | 56.3 | 53.8 | 55.2 | 48.3 | 44.3 | 45.7 |
| $\mathcal{F}$ (Mean) | 61.3 | 59.2 | 58.3 | 58.7 | 52.4 | 49.6 | 52.3 |

Figure 6(e). Table 2 (b)(e) demonstrate the contribution of this concentration regularization term, e.g., compared to (b), the $\mathcal{J}$ in (e) decrease by 8% without this regularization term.

**Orthogonal regularization.** The orthogonal regularization term enforces points to match back to themselves after a cycle of forward and backward transformation. As shown in Figure 6 (f), the model trained without the orthogonal regularization term is less effective in preserving local structures. The effectiveness of the orthogonal regularization is also validated quantitatively at Table 2 (e) and (f).

### 4.4 Tracking Pose Keypoint Propagation on the J-HMDB Dataset

We demonstrate that our model learns accurate correspondence by evaluating it on the J-HMDB dataset [19], which requires precise matching of points compared to the coarser propagation of masks. Given the 15 ground truth human pose key-points in the first frame, we propagate them to the remaining frames. We quantitatively evaluate per-

Table 3: Tracking results on OTB2015 [53]

| Model | Supervised | AUC score (%) |
|---|---|---|
| UDT [46] | × | 59.4 |
| Ours | × | 59.2 |
| ResNet-18 | ✓ | 55.6 |
| Fully Supervised [2] | ✓ | 58.2 |

formance using the probability of correct keypoint (PCK) metric [57], which measures the ratio of joints that fall within a threshold distance from the ground truth joint locations. We show quantitative evaluations against the state-of-the-art methods in Table 5 and qualitative propagation results in Figure 4(b). Our model performs well versus all self-supervised methods [52, 45] and notably achieves better results than ResNet-18 [13] trained with classification labels [9].

### 4.5 Visual Tracking on the OTB Dataset

Other than the tasks that require dense matching, e.g., segmentation or keypoints propagation, the features learned by our model can be applied to object matching tasks such as visual tracking, because of its capability of localizing an object or a relatively global region. Without any fine-tuning, we directly integrate our network trained via self-supervision into a classic tracking framework [46, 37] based on correlation filters, by replacing the Siamese network in [46, 37] with our model, while keeping other parts in the tracking framework unchanged. Even without training with a correlation filter, our features are general and robust enough to achieve comparable performance on the OTB2015 dataset [53] to methods trained with this filter [46], as shown in Table 3. Figure 4(d) shows that our learned features are robust against occlusion (left), object scale, as well as illumination changes (right) and can track objects through a long sequence (hundreds of frames in the OTB2015 dataset).

Table 4: Segmentation propagation on VIP [59].

| Model | Supervised | mIoU | $AP^r_{vol}$ |
|---|---|---|---|
| DeepCluster. [6] | × | 21.8 | 8.1 |
| Wang et al. [52] | × | 28.9 | 15.6 |
| Ours | × | 34.1 | 17.7 |
| ResNet-18 | ✓ | 31.8 | 12.6 |
| Fully Supervised [38] | ✓ | 37.9 | 24.1 |

Table 5: Kepoints propagation on J-HMDB [19].

| Model | Supervised | PCK@.1 | PCK@.2 |
|---|---|---|---|
| Vondrick et al. [45] | × | 45.2 | 69.6 |
| Wang et al. [52] | × | 57.3 | 78.1 |
| Ours | × | 58.6 | 79.8 |
| ResNet-18 | ✓ | 53.8 | 74.6 |
| Fully Supervised [56] | ✓ | 68.7 | 92.1 |

### 4.6 Semantic and Instance Propagation on the VIP Dataset

We evaluate our method on the VIP dataset [59], which includes dense human parts segmentation masks on both the *semantic* and *instance* levels. We use the same settings as Wang et al. [52] and resize the input frames to $560 \times 560$. For the semantic propagation task, we propagate the semantic

segmentation maps of human parts (e.g., arms and legs) and evaluate performance via the mean IoU metric. For the part instance propagation task, we propagate the instance-level segmentation of human parts (e.g., arms of the first person or legs of the second person) and evaluate performance via the mean average precision of the instance-level human parsing metric [26]. Table 4 shows that our method performs favourably against all self-supervised methods and notably the ResNet-18 model trained on ImageNet with classification labels for both tasks. Figure 4(c) shows sample semantic segmentation propagation results. Interestingly, our model correctly propagates each part mask onto an unseen instance (the woman which does not appear in the first frame) in the second example.

## 5    Conclusions

In this work, we propose to learn correspondences across video frames in a self-supervised manner. Our method jointly tackles region-level and pixel-level correspondence learning and allows them to facilitate each other through a shared inter-frame affinity matrix. Experimental results demonstrate the effectiveness of our approach versus the state-of-the-art self-supervised video correspondence learning methods, as well as supervised models such as the ResNet-18 trained on ImageNet with classification labels.

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
