[Supplementary Material]

# Joint-task Self-supervised Learning
# for Temporal Correspondence

**Xueting Li**[1]*, **Sifei Liu**[2]*, **Shalini De Mello**[2], **Xiaolong Wang**[3], **Jan Kautz**[2], **Ming-Hsuan Yang**[1]
[1]University of California, Merced, [2]NVIDIA, [3] Carnegie Mellon University

## 1 Texture Propagation

Figure 1: Texture Propagation.

In Figure 1, we show results of texture propagation. Following Wang et al. [2], we overlay a texture map on the object in the first video frame, then propagate this texture map across the rest of the video frames. As shown in Figure 1, our model is able to preserve the texture well during propagation, this indicates that our model is able to find precise correspondences between video frames.

## 2 Instance Segmentation Propagation on DAVIS-2017

In Figure 2, we show more instance mask propagation results on the DAVIS-2017 dataset [1]. Our model is resilient to rapid object shape and scale changes, e.g., the horse, the motorbike and the cart in Figure 2.

In Figure 3, we visualize the process of including the localization module during inference. Given the instance mask of the first frame, we first propagate each point (marked as green) from the reference frame to the target frame by localizing a bbox on it before matching. Instead of directly applying the center as described in Section 3.2 in the paper, we refine the center at inference by applying the mean-shift algorithm, i.e.,

$$C_t = \frac{\sum_{i=1}^{N} K(l_i - C_{t-1})l_i}{\sum_{i=1}^{N} K(l_i - C_{t-1})} \tag{1}$$

where $l_i$ is the coordinate of the $i^{th}$ pixel, the $C$ is the center of all $l_i$ at the $t^{th}$ iteration, and $K(a - b) = e^{\|a-b\|^2}$. Scale is estimated via Eq.(4) as well, see the bboxes in Figure 3. The green points in Figure 3 illustrate the individually propagated points and the red bounding box indicates the estimated bounding box of an object in the target frame. We then propagate the instance segmentation mask within the bounding box in the reference frame to the bounding box in the target frame.

---

Figure 2: Instance mask propagation results.

Figure 3: Visualization of the process of including the localization module during inference.