[Reviews · NeurIPS 2019]

Reviewer 1



The work does not include original ideas. It is exclusively a collection of previous ideas combined together in a rather classical way. Major remarks: Equation (6) makes loss non-smooth and non-differentiable. The authors do not discuss how they handle this. I assume they use the typical approach by getting the right 'case' in the forward step and then doing back-prop on the fixed smooth function. Line 129-130: Temperature should be tried to impose sparseness. In addition, the Gumble trick can be used here and should be tried. Minor remarks: Lines 218-220: They are unclear. The first part says that they pretrain the AE, but the next sentences says that they train without any pre-training. Line 120: that take

Reviewer 2



Self-supervised learning is an interesting topic to work on. This paper presents several constraints/losses to regularize the training of deep nets, including region-level localization, concentration regularization, and orthogonal regularization. The experimental results show that the proposed method outperforms the main competing baseline [49] (a CVPR'19 paper).

Reviewer 3



1) The paper is very well written and easy to follow 2) The approach being new, the intuition behind every model decision is well explained and it was easy to understand the model architecture design, the loss functions etc. 3) The paper thoroughly reviewed the previous work in tracking and fine-grained correspondence and clearly communicates the novelty of this paper. 4) The paper will be very significant as it's a new way to think about self-supervised learning in videos.

[Author Response · NeurIPS 2019]

Before addressing the issues raised by the reviewers, we present the results of using our self-supervised representation for segmentation mask propagation on the DAVIS dataset (Table 1), in comparison to a SOTA self-supervised method [49] and the ImageNet pre-trained representation.

Table 1: DAVIS-2017 segmentation results.

| Model | $\mathcal{J}$ (Mean) |
|---|---|
| Self-supervised, SOTA [49] | 43.0 |
| ImageNet Representation | 49.4 |
| Self-supervised, Ours | **57.7** |

We sincerely urge the area chair and reviewers to evaluate this paper in the context of advances accomplished by our proposed self-supervised learning method. This is the **first** paper to show large improvements over an ImageNet pre-trained representation. The contributions are significant, since the performance of the existing SOTA [49] is far below our method and also below that of the ImageNet pre-training method.

**Novelty and Contributions. (R1, R2)**  We would like to remind the reviewers and the area chair of the challenges and difficulties in leveraging various self-supervision signals. Most existing methods still focus solely on applying multiple losses, e.g., [17, 43, 44, 49], and achieve marginal improvements. Instead of performing multi-loss training, we propose a novel framework, in which we build one task upon another that progressively improve each other during training. The shared affinity matrix bridges these tasks, and facilitates iterative improvements. The proposed framework not only improves previous self-supervised approaches significantly, but outperforms supervised learning with human supervision by a large margin. These contributions are significant in the field of self-supervised learning.

The contributions of this work are also demonstrated by our ablation study, i.e., Table 2 in the paper. All the proposed components, e.g., coarse localization, concentration and orthogonal regularization, contribute to the performance gain. We note that these components are novel and have not been explored in prior work.

**Similarity to [49] (R2)**  The key ideas of our paper are not similar to [49]: (i) [49] matches patches instead of pixels, and the entire fine-grained matching part is missing; (ii) [49] models the locations independently via a STN, while we track both features and locations uniformly via the same affinity matrix; (iii) The cycle-consistency in our method refers to the orthogonal regularization of the affinity matrix, which is very much different from the cycle-consistency tracking loss in [49]. The achieved performance gain of this work comes from all the above-mentioned algorithmic components and concentration regularization, rather than engineering work.

Table 2: Comparison against optical flow methods.

| Model | $\mathcal{J}$ (Mean) | $\mathcal{F}$ (Mean) |
|---|---|---|
| FlowNet2 [16] | 26.7 | 25.2 |
| PWC-Net [39] | 35.2 | 37.4 |
| Ours | **57.7** | **60.0** |

Table 3: Ablation study on temperature (no track in testing).

| Temperature | $\mathcal{J}$ (Mean) | $\mathcal{F}$ (Mean) |
|---|---|---|
| 1 | 56.8 | 59.5 |
| 2 | 52.3 | 56.2 |
| 10 | 51.0 | 55.5 |

**Which methods should the work compare with? (R2)**  We note that the focus of this work is not on learning unsupervised flow. Rather, it aims to achieve the same goal as presented in [43] and [49] – learning unsupervised correspondences. These two tasks are significantly different: one focuses on learning a feature matching network that can track regions and pixels between frames over a long period, while the other emphasizes modeling subtle displacements between adjacent frames. Taking Table 2 as an example, for the segmentation propagation task on the DAVIS dataset, even the SOTA **fully-supervised** flow methods perform much worse than any of the unsupervised correspondence methods. All the latest methods related to unsupervised video correspondences are evaluated, including several concurrent works, e.g., [7,8] in the supplementary material. The comparisons with UDT [44] are thoroughly discussed in Section 4.4.

In the following, we address the other comments by reviewers.

**Sharpness in the Softmax layer. (R1)**  We experimented with various temperatures in the softmax function and found that setting it to 1 achieves the best results. See Table 3.

**Equation (6) makes loss non-smooth and non-differentiable. (R1)**  With Eq. (6), we only assign a penalty to pixels that move outside the bounding box, which is achieved by multiplying a rectangular mask to the loss term. In practice, there is no need to specify gradients on the boundary to enforce smoothness. We will clarify this in the revised paper.

**Scale estimation in equation 4. (R3)**  We compute the scale strictly according to Eq. (4), rather than computing the maximum distance. Please refer to Figure 3 in the supplementary for visualizations of the estimated scales.

**Which dataset to train the auto-encoder? (R3)**  The MSCOCO dataset is large and diverse for training an image auto-encoder, e.g., with a shallow 6-layer encoder. The effectiveness of our trained auto-encoder is also validated in Table 2 (g). Under the same settings, we achieve 11.1% higher accuracy than [43] that does not use the auto-encoder.

**Evaluating learned representations on the image based tasks (R3)**  Transferring the learned representations from video correspondence to image based task is certainly interesting. We will explore it in the future.

[Meta-Review · NeurIPS 2019]

The paper presents a new approach to tracking and pixel level correspondence using self-supervised learning in video. It goes in the direction of multi-task learning. As well results are solid. The reviewers at the beginning gave a score of 5,6 and 7, than after rebuttal also the more skeptic reviewer was convinced to improve its rate. . Even the novelty is not to high, the area chair and reviewers agree for an acceptance